# Switching from Sugar- to Artificially-Sweetened Beverages: A 12-Week Trial

**DOI:** 10.3390/nu15092191

**Published:** 2023-05-04

**Authors:** Michael D. Kendig, Julie Y. L. Chow, Sarah I. Martire, Kieron B. Rooney, Robert A. Boakes

**Affiliations:** 1School of Life Sciences, University of Technology Sydney, Sydney, NSW 2007, Australia; 2School of Psychology, UNSW Sydney, Sydney, NSW 2052, Australia; 3School of Psychology, University of Sydney, Sydney, NSW 2006, Australia; 4Discipline of Exercise and Sport Science, Faculty of Medicine and Health, University of Sydney, Sydney, NSW 2006, Australia

**Keywords:** sugar-sweetened beverages, short-term memory, sweet taste preference, artificially sweetened beverages, metabolic health

## Abstract

Background: Consumption of sugar-sweetened beverages (SSBs) forms the primary source of added sugar intake and can increase the risk of metabolic disease. Evidence from studies in humans and rodents also indicates that consumption of SSBs can impair performance on cognitive tests, but that removing SSB access can ameliorate these effects. Methods: The present study used an unblinded 3-group parallel design to assess the effects of a 12-week intervention in which young healthy adults (mean age = 22.85, SD = 3.89; mean BMI: 23.2, SD = 3.6) who regularly consumed SSBs were instructed to replace SSB intake with artificially-sweetened beverages (*n* = 28) or water (*n* = 25), or (c) to continue SSB intake (*n* = 27). Results: No significant group differences were observed in short-term verbal memory on the Logical Memory test or the ratio of waist circumference to height (primary outcomes), nor in secondary measures of effect, impulsivity, adiposity, or glucose tolerance. One notable change was a significant reduction in liking for strong sucrose solutions in participants who switched to water. Switching from SSBs to ‘diet’ drinks or water had no detectable impact on cognitive or metabolic health over the relatively short time frame studied here. This study was prospectively registered with the Australian New Zealand Clinical Trials Registry (ACTRN12615001004550; Universal Trial Number: U1111-1170-4543).

## 1. Introduction

Intake of sugar-sweetened beverages (SSBs) comprises the major source of added sugar consumption in youth and adults [1] and has been associated with an increased risk of developing metabolic and cardiovascular disease [2,3,4,5]. SSBs are less satiating than sugar presented in solid foods [6] and are now available as a diverse range of products; for example, carbonated soft drinks, energy drinks, iced coffees and teas, flavoured milks, juice drinks and ‘bubble’ teas, among other products. Increasing evidence for adverse effects associated with SSB intake has generated policy changes, most notably the implementation of taxes on SSBs that recent meta-analyses indicate have been successful in reducing purchase and consumption [7,8].

SSB intake has also been associated with an increased risk of cognitive impairment in meta-analyses of epidemiological data [9,10] and reviews of rodent studies [11], forming part of a larger evidence base showing adverse cognitive effects of diets high in sugar and/or fat on hippocampal function and spatial learning and memory in rodents [12] and humans [13]. Notably, short-term exposure to high-sugar, high-fat diets can impair hippocampal-dependent forms of cognition within a week [14,15], suggesting that dietary effects on cognition are not contingent on changes in adiposity or body weight. Evidence from rodent studies indicates that the cognitive impairments associated with SSB intake can be ameliorated by replacing the sugar solution with water or a non-nutritive sweetener [16]. Beneficial cognitive effects have been reported when animals fed a high-fat, high-sugar ‘cafeteria’-style diet are switched to regular chow [17,18] and improvements are not contingent on reductions in body weight or fat mass [19]. However, whether interventions that reduce SSB intake can improve hippocampal-dependent forms of cognition in humans remains unclear. This formed the primary aim of the present study.

Interventions designed to reduce SSB intake can improve metabolic parameters, as shown in meta-analyses of randomised clinical trials of studies in adults with overweight/obesity or at risk of diabetes [20] and systematic reviews of studies in children [21]. In a study of adolescents with overweight and obesity, an intervention designed to minimise SSB intake resulted in a smaller BMI increase over 12 months relative to controls [22]. Similarly, an analysis of beverage intake trends in adults with prehypertension or stage 1 hypertension in the PREMIER trial found that reductions in SSB intake were associated with greater weight loss over 18 months, even though dietary advice did not explicitly recommend reducing calories from sugary drinks [23]. In a recent randomised controlled trial, however, Ebbeling and colleagues found that switching adult SSB consumers to artificially sweetened or unsweetened beverages for 12 months did not significantly reduce body weight or the ratio of serum triglyceride to high-density lipoprotein cholesterol, relative to a group maintained on their usual SSB intake [24]. Changes in BMI, blood glucose and other metabolic parameters were among the secondary outcomes assessed in the present study.

It is unclear whether switching from SSBs to low-calorie or unsweetened alternatives leads to changes in other components of the diet, such as a preference for sweet foods. Some evidence indicates that replacing caloric beverages with ‘diet’ options or water can improve overall dietary patterns [25]. A systematic review of 21 studies found some evidence that higher exposure to sweet tastes in the diet reduced preference for sweet tastes over the short term, albeit from a small and heterogeneous literature [26]. In a recent clinical trial, preferred sweetness concentration reduced significantly from baseline in participants who switched from SSBs to low-calorie beverages or water, but not those maintained on SSBs [24]. Here we assessed changes in liking ratings for five concentrations of sucrose solution before and after the SSB replacement intervention.

Thus, the present study used an unblinded, 3-group parallel design to examine changes in cognitive, metabolic, and affective parameters across a 12-week intervention in young healthy adult SSB consumers who were asked to replace their regular SSB intake with non-nutritive ‘diet’ drinks (Diet group) or water (Water group); or to continue consuming SSBs in the form of carbonated soft drinks (Sugar group). Short-term memory and waist circumference (primary outcomes) were assessed at baseline, weeks 6 and 12 of the 12-week intervention, and 12 weeks after the intervention. Adiposity, blood glucose, and lipids, sweet taste preference, and affective measures were among the secondary outcomes assessed before and after the intervention.

## 2. Method 

### 2.1. Ethics and Protocol Registration 

This trial was prospectively registered with Sydney Local Health District Clinical Trials (Protocol No X14-0366) and the Australian New Zealand Clinical Trials Registry (ACTRN12615001004550; Universal Trial Number: U1111-1170-4543) and was conducted according to the guidelines set out in the Declaration of Helsinki. Procedures were approved by the University of Sydney Human Research and Ethics Committee (Project 2015/074). All participants provided informed consent. 

### 2.2. Participants 

Recruitment was conducted from April 2016 to September 2018, eligibility criteria included aged between 18 and 35 years; with a BMI between 17.5 and 30. Four participants had a BMI < 18.5 (*n* = 2, 1, 1 in Sugar, Diet, and Water groups, respectively). Participants were otherwise healthy and self-reported regularly drinking at least 2 L of sugar-sweetened drinks per week. The study was open to both males and females. In a prior Australian study using a convenience sample, Francis and Stevenson [27] found that a self-reported high-fat, high-sugar diet reduced performance on the Logical Memory test relative to a self-reported low dietary Saturated fat and refined sugar intake LFS diet with an effect size of d = 0.81. To detect the effects of our diet intervention with 80% power, a more conservative effect size of *d* = 0.70, and with a = 0.05 (two-tailed), a sample size of 34 per group was required. We anticipated an attrition rate of 33% based on a previous trial of SSB intake [28]. Therefore, we sought to recruit and randomise a total of 153 participants to achieve a final sample of 34 per group.

### 2.3. Design 

This study used an unblinded 3-group (Sugar, Diet, and Water) parallel design involving block randomisation for sets of 3 participants, such that we attempted to match the groups on BMI and baseline intake of sugar-sweetened beverages. For each week of the 12-week intervention, participants in the Sugar group received 4.5 L of sugar-sweetened beverages (as 12 × 375 mL cans); the Diet group received 4.5 L of artificially-sweetened beverages (as 12 × 375 mL cans); and the Water group received 4.5 L bottled water (as 9 × 500 mL bottles). Sugar and Diet group participants could select from a range of commercially available carbonated beverages. The sweeteners in the diet drinks included aspartame, acesulfame-K, and sucralose. All participants were asked to abstain from sweetened drinks beyond those provided; participants in the Sugar and Diet groups were instructed to consume no more than 3 cans per day. Adherence was assessed at weekly drink collections via a brief compliance questionnaire. There were no further dietary restrictions.

### 2.4. Primary and Secondary Outcomes 

The primary outcomes were changes in cognition on the Logical Memory test (LM) and in waist circumference to height ratio (WC:H; [29]), with each assessed at weeks 1, 6, and 12 of the intervention, and at follow-up (~12 weeks after the intervention). The study was powered based on expected changes in the primary cognitive outcome. Secondary outcomes included the Controlled Oral Word Association Test (COWAT), BMI, blood pressure, body fat measured by bioelectrical impedance analysis, blood glucose, lipids and uric acid, and sucrose preference.

In the LM test, a subtest of the Wechsler Memory Scale—Revised [30], participants are read two unique stories each containing 25 semantic items. Verbal recall is tested immediately and after a 20–30 min delay; the percentage of items retained at the delayed test is taken as a measure of memory. Four pairs of unique stories were used in a counterbalanced order across tests, incorporating alternative test stories developed by Sullivan [31] and Schnabel [32] to the original pair [30]. Test sessions were recorded and scored live by the experimenter. Although blinding was not possible at this stage, audio recordings were not identifiable by the treatment group, allowing for half the data to be re-scored at a later date by other team members not involved in test administration who were unaware of intervention group. These scores were closely correlated with the original data (*r* = 0.96). In the few tests where scores differed by more than 3 SDs, data were re-scored by a third scorer and this final score was used in analyses.

In the COWAT [33,34], participants were asked to produce as many unique words as possible that begin with a given letter in 60-s, for three different letters, excluding proper nouns, plurals, and word derivations. Here, three different letter sets of equivalent difficulty were used in Tests 1–3 (F-A-S, C-F-L, and P-R-W), with order counterbalanced [34,35]. At Test 4 we used a fourth letter set (T-D-M) chosen to match the other sets in terms of the mean number of associations and the Thorndike-Lorge score count (see Table 1 in ref. [36]). The total words generated for the three letters was used as a verbal fluency measure, and word generation in 15 s bins was also examined to explore spontaneous word production over time.

To estimate SSB intake, at each test participants completed the 15-item Brief Beverage Intake Questionnaire (BEVQ-15; [37]), modified to capture sugary drinks commonly consumed in Australia, including carbonated soft drinks, flavoured milks, sweetened teas, and fruit juices/cordials. To aid recall and accurate estimation, the questionnaire was completed with a visual aid depicting common beverage serving sizes in millilitres. Participants’ estimated frequency and volume were then converted to an average weekly estimate of SSB intake (ml). Participants also completed the Dietary Fat and Free Sugar—Short Questionnaire (DFSQ; [38]), a validated 26-item questionnaire assessing the consumption of high-fat, high-sugar foods. While this questionnaire is typically applied to the last year, when administered during the intervention we instructed participants to estimate their intake of each item since the last measure (i.e., over the preceding 6 weeks). At each test session, participants also completed the 30-item Revised Barratt Impulsivity Scale—Short Form (BIS-11; [39,40]), and the Depression, Anxiety, and Stress Scale (DASS-21), which is a 21-item measure of depression, anxiety and stress validated against the original 42-item questionnaire [41,42]. At preliminary screening and follow-up, information on sleep, physical activity, and medical history was collected.

### 2.5. Procedure 

Recruitment and Screening. The study was advertised via the University of Sydney research volunteer website, social media, and flyers distributed around campus. Prospective participants were sent information about the study and contacted for a brief phone interview to estimate weekly SSB intake (>2 L/week) and confirm age (18–35) and no health-related issues. Those meeting these criteria were invited to an in-person screening session, where the BEVQ-15, DFSQ, and BIOFORM questionnaires were administered, and weight and height were recorded. Participants were excluded if BEVQ-15 results indicated that weekly SSB intake was <2 L or if the participant reported poor health or medical conditions indicated on the BIOFORM (e.g., diabetes, heart problems, psychoses). Eligible participants were scheduled to return to the laboratory the following week to begin the intervention (Test 1). Participants were pseudo-randomly allocated to Sugar, Diet, or Water groups, matching on SSB intake (obtained from BEVQ-15) and BMI at screening.

Intervention (12 weeks). Participants attended the laboratory for tests at baseline (Test 1), weeks 6 (Test 2), and 12 (Test 3) of the intervention, and at follow-up 12 weeks after the intervention (Test 4). At each Test, participants completed the LM and COWAT cognitive tests, and BEVQ-15, DFSQ, BIS-11, and DASS-21 questionnaires. Weight, height, waist circumference, blood pressure, and heart rate were measured at rest, and body fat and skeletal muscle were quantified by Bio-electrical Impedance Analysis (BIA) using a Quantum IV machine (RJL systems, Clinton Township, MI, USA). 

Additional measures of sweet taste preference and glucose tolerance were collected at Tests 1 and 3. Participants were instructed to record their food intake, abstain from excessive exercise and alcohol, and fast from 11 pm the night before (water was permitted) for a shortened oral glucose tolerance test. After collecting fasted measures of blood glucose (Accuchek^®^, uric acid, and blood lipids (CardioChek^®^), participants consumed a 300 mL glucose solution (Carbotest^®^; 50 g glucose), with blood glucose and uric acid measures taken 15, 30, 45 and 60 min later, during which time participants completed cognitive tests and questionnaires. After the OGTT, participants completed a sucrose preference test in which they tasted 1.5%, 3%, 6%, 12%, and 24% sucrose solutions and rated their liking for each concentration on a 140mm visual analogue scale (anchors: dislike very much and like very much). Solutions were tasted and rated in an ascending and then descending order of concentration, with the average of the two ratings calculated to identify participants’ most-preferred solution. Participants were instructed to rinse with water between each concentration.

At the end of Test 1, participants were told their intervention group and given drinks for the first week. Thereafter, participants visited the laboratory weekly to collect drinks and complete a compliance questionnaire to confirm intake of the prescribed drinks and check no other SSBs were consumed. The structure of Tests 2 and 4 was identical except that the OGTT and sucrose preference tests were not administered. After Test 4 (Follow up), participants were emailed a summary of their metabolic data and a letter thanking them for their participation. In total, participants were compensated $230AUD: $20 after screening (paid regardless of eligibility), $40 following Test 1, $50 after Test 2; and $60 after Test 3 (end of intervention) and Test 4 (Follow-up). 

### 2.6. Statistical Analysis 

Data were analysed for participants that completed the first three tests using a per-protocol approach. This approach was chosen as we sought to investigate the cognitive and physiological effects of reducing SSB intake; thus, participants who continued to consume SSBs were excluded. To assess this, we analysed participants’ sugar-sweetened beverage intake, which was estimated with the Hedricks BEVQ questionnaire at weekly check-in sessions. Blinding was not possible for the test administrators, but primary outcome data were re-scored by team members unaware of the intervention group. Raw data were entered into a master spreadsheet with treatment intervention coded until the completion of the per-protocol review and statistical analyses. We excluded 17 participants whose average weekly SSB intake was more than 1.5 times their group’s interquartile range, as shown in Appendix A, leaving group sizes of *n* = 27 (Sugar), *n* = 28 (Diet) and *n* = 25 (Water). In these participants, the mean reported SSB intake at the baseline screening session was 6.7 L/week (SD = 3.7 L). 

Group differences in primary outcomes (LM retention score and waist circumference:height) were tested by Analysis of Covariance (ANCOVA) including group and gender as factors and controlling for baseline measures of BMI, BEVQ-15, and DFSQ scores. Secondary outcomes were analysed similarly. Follow-up data (Test 4) were analysed in ANCOVAs that compared differences between groups and gender, controlling for performance at Test 3, BMI at Test 3, change in BEVQ score from Test 3 to Test 4, and DFSQ-Sugar subscale score at Test 4. Greenhouse-Geisser corrections were applied where Mauchly’s Test of Sphericity was significant.

## 3. Results 

### 3.1. Recruitment

A flowchart of participant recruitment and attrition is shown in Appendix A. Of the 118 participants allocated to the three intervention groups, 97 completed the 12-week intervention (39 females, M_age_ = 22.85, SD = 3.89; M_BMI_: 23.2, SD = 3.6). Of these, 87 returned for a follow-up test session 12 weeks after Test 3.

### 3.2. Primary Outcomes: Logical Memory

Figure 1 presents estimated marginal means of the memory retention score (i.e., delayed recall as a percentage of immediate recall) across tests. The group × test interaction was not statistically significant (*F*(4,142) = 0.516, *p* = 0.724, η_p_^2^ = 0.014) and there were no other significant main or interaction effects (largest *F* = 2.02). Similarly, at follow-up, ANCOVAs found no significant effects of Group or Sex (nor their interaction) on retention scores (largest *F* = 2.42, *p* = 0.097, η_p_^2^ = 0.070). Overall raw scores for the story pairs (out of 50) did not differ between groups and did not vary across tests (immediate and delayed recall [SEM] at Test 1: 21.5 [0.77], 19.7 [0.77], Test 2: 25.6 [0.75], 24.1 [0.74], Test 3: 24.7 [0.90], 22.8 [0.89]; Test 4: 23.9 [1.11], 22.0 [1.02]).

### 3.3. COWAT

Figure 2 shows the total words generated in the COWAT in Tests 1–4. The group × test interaction for performance in Tests 1–3 was not significant (*F*(4, 142) = 1.53, *p* = 0.197, η_p_^2^ = 0.041) and there were no significant group differences at follow-up, analysed in a mixed-model ANCOVA that included Test 3 performance as a covariate (*F*(6,192) = 1.52, *p* = 0.173, η_p_^2^ = 0.045). 

### 3.4. Affective Measures: DASS-21

As shown in Figure 3, there were no significant group × test interactions for total DASS-21 scores (*F*(4, 142) = 0.446, *p* = 0.775, η_p_^2^ = 0.012) nor on the depression, anxiety or stress subscales (largest *F*(4, 142) = 1.82, *p* = 0.127, η_p_^2^ = 0.049). Similarly, an ANCOVA of DASS-21 scores at follow-up found no significant main or interaction effects (largest *F* = 1.31).

### 3.5. Affective Measures: Impulsivity (BIS-11)

The group × test interaction was not significant for total BIS scores (Figure 4A; *F*(2, 70) = 2.25, *p* = 0.113, η_p_^2^ = 0.060), motor (Figure 4C; *F*(2, 70) = 0.284, *p* = 0.753, η_p_^2^ = 0.008) or attention (Figure 4D; *F*(2, 70) = 0.339, *p* = 0.714, η_p_^2^ = 0.010) subscales. However, the group × test interaction was significant for the non-planning subscale (Figure 4B; *F*(2,70) = 6.73, *p* = 0.002, η_p_^2^ = 0.161). Exploratory follow-up tests identified a significant increase only in the Diet group over tests (Mean difference = 2.06, *p* = 0.002; 95% CI [0.82, 3.31]). There were no significant main or interaction effects on total BIS-11 scores or the individual subscales at follow-up (largest *F* = 1.48).

### 3.6. Primary Outcome: Waist Circumference to Height Ratio 

Figure 5 shows changes in the WC:H ratio across groups from the baseline to the end of the intervention. An ANCOVA found no significant interaction between group and test (*F*(4, 142) = 0.408, *p* = 0.803, η_p_^2^ = 0.011) and no other significant main or interaction effects involving group, sex, or their interactions (largest *F* = 1.09). At follow-up (*n* = 67), there were no significant main effects of Group or Sex, and no interaction between the two factors (largest *F*(2, 65) = 1.26, *p* = 0.291). 

### 3.7. OGTT

Figure 6 shows changes in blood glucose across the 60-min test at baseline (Test 1) and after the intervention (Test 3) for each group. A repeated-measures ANCOVA showed that the key interaction between group, test (Test 1 vs. Test 3), and time (0–60 min) was not significant (*F*(8, 284) = 0.832, *p* = 0.575, η_p_^2^ = 0.023). Uric acid levels did not vary between groups or across tests (see Appendix A).

### 3.8. Metabolic and Physiological Measures

The group × test interactions were not significant for percent body fat (Figure 7A; *F*(3.36, 119.4) = 0.457, *p* = 0.735) and skeletal muscle (Figure 7B; *F*(3.19, 113.3) = 1.12, *p* = 0.35). At follow-up (Test 4), however, there were significant group differences in percent body fat (*F*(2,64) = 3.82, *p* = 0.027, η_p_^2^ = 0.105) and skeletal muscle (*F*(2,64) = 4.50, *p* = 0.015, η_p_^2^ = 0.123). Exploratory post-hoc contrasts indicated that these effects were driven by differences between the Diet and Water groups, with the Diet group exhibiting higher fat mass (Mean difference = 2.72%; *p* = 0.009; 95% CI [0.71, 4.73]) and lower skeletal muscle mass (Mean difference = −1.56%; *p* = 0.003; 95% CI [−2.56, −0.56]). There were no significant group × test interactions for systolic blood pressure, fasting blood lipid parameters, BMI, or body weight (largest *F*(4, 142) = 1.21, *p* = 0.311, η_p_^2^ = 0.033). It should be noted that the number of data points varied for blood lipid measures due to the detection threshold on the *CardioChek* blood testing device (*N* = 75 for cholesterol, 74 for HDL, 68 for LDL, 74 for triglycerides). 

### 3.9. SSB Intake (BEVQ)

Self-reported SSB intakes, as measured by the BEVQ, are shown in Figure 8. A repeated-measures ANOVA on SSB intakes across Tests 1–3 found a significant test × group interaction (*F*(2.36, 84) = 23.9, *p* < 0.001, η_p_^2^ = 0.40), driven by significantly higher SSB intakes in the Sugar group relative to the Diet and Water groups at Test 2 and Test 3 (both *p* < 0.001). At follow-up (Test 4), SSB intake did not differ as a function of prior intervention groups (*F*(2,67) = 0.23, *p* = 0.80, η_p_^2^ = 0.007).

### 3.10. DFSQ

The group × test interaction for total DFSQ scores (Figure 9A) was not significant (*F*(4, 144) = 0.336, *p* = 0.854, η_p_^2^ = 0.009). Notably, however, there was a significant quadratic trend for ‘Test’ (*F*(2,144) = 3.88, *p* = 0.023, η_p_^2^ = 0.051), reflecting that scores decreased in all groups from Test 1 to Test 2 and were stable from Tests 2 to 3. For the sugar subscale of the DFSQ (Figure 9B) the group × test interaction was significant (*F*(3.53, 127) = 4.97, *p* = 0.002, η_p_^2^ = 0.121), reflecting the larger decrease in sugar scores in Diet and Water groups relative to the Sugar group at Tests 2 and 3 (both *p* < 0.001). The group × test interaction for scores on the fat subscale of the DFSQ (Figure 9C) was not significant (*F*(4, 144) = 1.353, *p* = 0.253, η_p_^2^ = 0.036). At follow-up (Test 4), there were no group differences in total DFSQ scores (*F*(2, 74) = 0.372, *p* = 0.691, η_p_^2^ = 0.011) or on the sugar (*F*(2, 74) = 0.382, *p* = 0.684, η_p_^2^ = 0.012) or fat (*F*(2, 74) = 0.727, *p* = 0.487, η_p_^2^ = 0.022) subscales.

### 3.11. Sweet Taste Preference

VAS ratings for the five sucrose concentrations at Test 1 and Test 3 are shown in Figure 10. The changes in liking ratings for each sucrose concentration (Test 3–Test 1) were analysed in a mixed-ANOVA with Group and Sex as between-subjects factors. The group × concentration interaction was significant (*F*(8, 296) = 2.57, *p* = 0.010, η_p_^2^ = 0.065) and there were no other significant main or interaction effects (largest *F*(1, 74) = 2.06, *p* = 0.155, η_p_^2^ = 0.027). To determine the source of this interaction, post-hoc pairwise comparisons were run at each concentration. These identified that liking decreased significantly more in the Water group than in the Diet group for both 12% sucrose (*p* = 0.020) and 24% sucrose (*p* = 0.032). Liking of 12% sucrose also decreased significantly more in the Sugar group than the Diet group (*p* = 0.028). 

## 4. Discussion

The present study tested the effects of replacing habitual SSB intake with either artificially-sweetened ‘diet’ drinks or water on cognitive, metabolic, affective, and taste preference measures across a 12-week intervention in young healthy adults. There were no significant differences between groups in primary outcomes of short-term memory (Logical Memory recall) and metabolic health (waist circumference/height ratio) nor in secondary outcomes of verbal fluency, impulsivity, affect, adiposity, and glucose tolerance, among others. There was some evidence that switching from sucrose to water reduced liking of concentrated sucrose solutions relative to a switch to diet drinks. Below we discuss the implications of these results, the potential shortcomings of the design, and the future directions arising from this work.

Performance on the Logical Memory test, a measure of hippocampal-dependent verbal memory and our primary cognitive outcome did not differ significantly between groups nor across the 12-week intervention. The absence of changes over time suggests that the use of four distinct story pairs was effective in minimising practice effects. Delayed recall percentages were consistently high, with participants reciting around 90% of the details they recalled at the immediate test. These values are consistent with previous studies (e.g., [32]), but raise the possibility that ceiling effects reduced the capacity to detect the effects of our intervention, suggesting that future work should incorporate more challenging test batteries. Similarly, no group differences were observed on the COWAT, a measure of verbal fluency, at any test. 

A consequence of the need to intersperse cognitive and metabolic tests in Tests 1 and 3 was that the acute and delayed Logical Memory tests, and the COWAT, were administered 30–60 min after consumption of glucose as part of the modified OGTT. A recent meta-analysis concluded that acute glucose consumption improves immediate verbal recall [43], raising the interesting possibility that prior glucose ingestion led to acute facilitation of performance on these tests, masking group differences. However, this appears unlikely given that raw LM scores on Tests 1 and 3—when glucose was consumed—were somewhat lower than on Test 2 when glucose was not consumed. More broadly, an interesting aspect of these data was that raw recall scores (as opposed to the percentage measure used in analyses) were 10–15% lower than means reported previously [32], albeit with high inter-individual variability. This difference in overall recall appeared specific to the LM test, since group means for word recall in the COWAT were consistently greater than norms reported in previous research, at least for the [F-A-S] letter set [44].

Regular assessment of SSB intake via the BEVQ questionnaire and weekly compliance checks at drink collections each suggested that the 12-week intervention was tolerable and enabled compliance by most participants. Notably, SSB intake did not differ significantly between groups at follow-up. Although variable within groups, BEVQ data suggested that mean SSB intake was ~3 L/week at follow-up, suggesting that participants in Diet and Water groups resumed SSB intake after the intervention, whereas participants in the Sugar group reduced their intake relative to the intervention conditions of 4.5 L/week. Nonetheless, in all groups, this represents a significant reduction relative to estimates taken prior to the intervention (~6 L/week). While recent evidence suggests that consumption of artificially-sweetened beverages may increase the risk of harmful metabolic and cardiovascular effects [45,46,47], none were observed on the measures included here, albeit over a relatively short 12-week time-frame.

Although our intervention did not involve any diet modifications other than to SSB consumption, DFSQ scores revealed a significant reduction in the self-reported frequency of fat and sugar intake at Tests 2 and 3. Reductions in scores on both fat and sugar subscales appeared to contribute to this result, with a greater decline in sugar scores in Diet and Water groups—consistent with the intervention instructions. Since questions about SSB intake form only a minority of items on the sugar subscale, these results suggest that participants also reduced their consumption of sugary solid foods. Notably, however, baseline DFSQ scores (M = 68.2) were higher than previous studies in comparable Australian samples of young adults (scores typically in the 50 s; [14,15,38]) and remained around the score of 57 suggested as the threshold for identifying a ‘poor’ diet not in line with the Australian Guide to Healthy Eating [38]. This may suggest that the apparently high intake of a Western-style diet obscured any potential benefits of minimising SSB intake over the 12-week intervention.

A broader limitation is that the present study was likely underpowered to detect significant group differences on these cognitive tests. Although our sample initially approached the necessary size (34/group) to detect significant group differences, identified by our power calculations, attrition at all stages of the intervention exceeded our estimates and financial constraints ultimately prevented continued recruitment. Sample size in analyses was further reduced by our decision to remove participants whose self-reported SSB intakes were outliers for their group at Tests 2 and 3. While this reduced power, we opted for this per-protocol approach to increase confidence that analyses focused on participants who complied with the study requirements. It is also possible, however, that SSB intake estimates via the BEVQ were subject to demand characteristics and that some participants retained in analyses were not compliant, even though participants were encouraged to complete the questionnaire honestly. Secondary measures of affect recorded at each test indicated few group differences in mood or impulsivity over time.

The 12-week SSB intervention had few effects on metabolic measures such as BMI, glucose tolerance, and blood lipids. Overall, this pattern of results aligns with the findings of a larger, 12-month clinical trial in which substituting artificially-sweetened or unsweetened beverages for sugar-sweetened beverages had no significant effects on cardiometabolic risk factors [24]. In the present study, body fat content appeared to decline from the beginning to the mid-point of the intervention, and was stable thereafter, but this overall effect did not differ between groups. This might be explained by the fact that the volume of weekly SSB intake prescribed for the Sugar group was below estimated consumption at baseline (4.5 L vs. ~6 L/week), meaning that all groups appeared to reduce SSB intake, promoting a loss in body fat. The finding that percent body fat was greater in the Diet vs. Water group at follow-up should be explored further with more extensive measures of habitual diet, as there was no clear dietary correlate of this result in terms of DFSQ or BEVQ scores.

There were interesting changes in sweet taste preference in participants who switched from SSBs to water, with this group showing a significant reduction in liking ratings for 12% and 24% sucrose relative to participants in the Diet group. Although this result suggests that the Water group’s reduced exposure to sweet taste (in the form of SSB intake) enhanced sensitivity to sweetness and shifted preference to a lower threshold, this interpretation is complicated by the fact that liking ratings for 12% sucrose also reduced significantly in the Sugar group relative to the Diet group. Our results are interesting to compare with those of Ebbeling et al. [24], who reported significant within-group reductions in preferred sweetness concentration in habitual SSB consumers who switched to unsweetened or artificially-sweetened beverages. By contrast, the Diet group in the present study appeared to increase liking for more concentrated solutions. Two minor differences in the protocol are that unlike Ebbeling et al. [24], our VAS anchors were in terms of liking (not perceived sweetness) and we did not explicitly ask participants to nominate their favourite concentration. Thus, further work is needed to clarify the effects on measures of sweet taste, given the heterogeneous evidence base to date [26]. In particular, it will be interesting to identify how effects obtained under controlled laboratory test conditions (such as in RCTs), might influence decisions related to the purchase of sweetened beverages or sweet foods in day-to-day scenarios, especially in instances where foods are signalled by distinct packaging. For example, a study of fourteen habitual SSB consumers who switched to artificially-sweetened beverages for 3 months found reductions in prefrontal cortex activity in response to high-fat, high-sugar foods, despite no overall change in body weight and no change in reported ‘liking’ of these foods [48].

In summary, we did not detect any effects of switching to ‘diet’ beverages or water on measures of short-term verbal memory or metabolic health in young healthy SSB consumers over a 12-week intervention. Results suggest that longer intervention periods may be necessary to observe improvements in these measures. Future work should explore whether the observed changes in sweet taste preference by Water group participants are robust in other situations.

## Figures and Tables

**Figure 1 nutrients-15-02191-f001:**
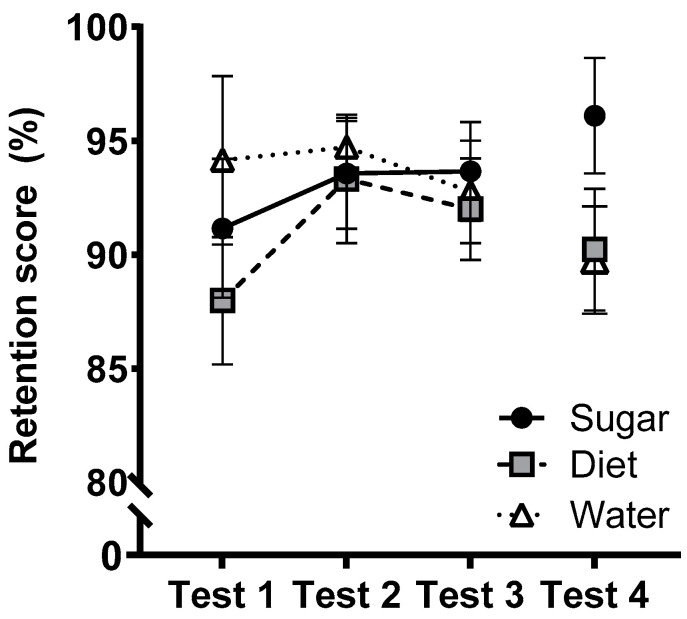
Estimated marginal means (±SE) of logical memory test retention scores (delayed recall as a percentage of immediate recall) across the 12-week intervention (Test 1–Test 3) and at follow-up (Test 4) for Sugar (*n* = 27), Diet (*n* = 28) and Water (*n* = 25) groups. At follow-up, *n* = 25 (Sugar), 27 (Diet), 24 (Water).

**Figure 2 nutrients-15-02191-f002:**
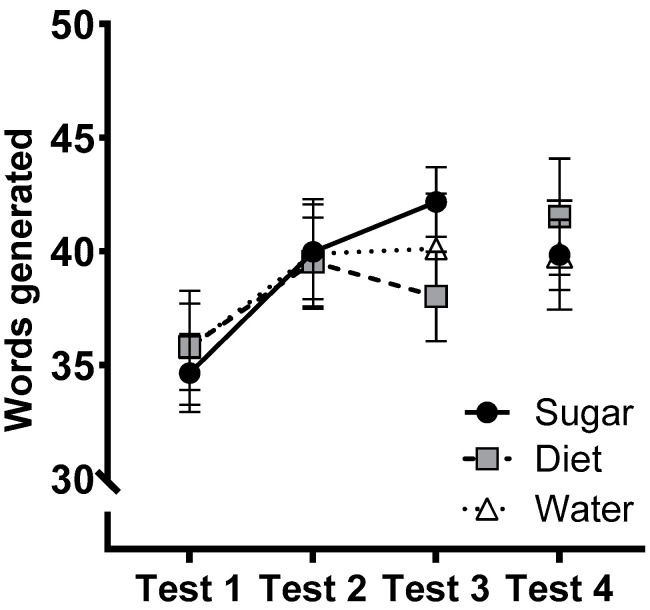
Estimated marginal means of the total number of words generated (±SE) across test sessions for participants in the Sugar (*n* = 27), Diet (*n* = 28), and Water (*n* = 25) groups. At follow-up, *n* = 25 (Sugar), 27 (Diet), 24 (Water).

**Figure 3 nutrients-15-02191-f003:**
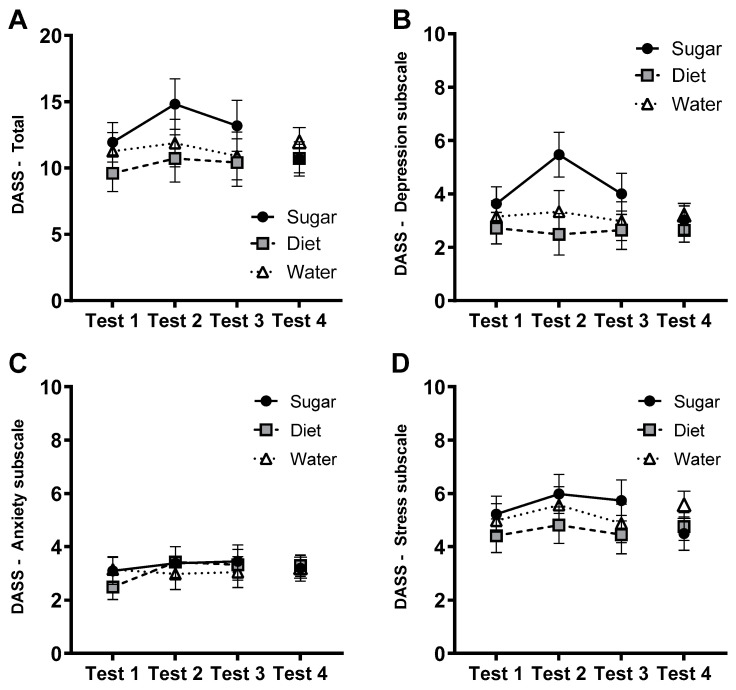
Estimated marginal means (±SE) of total DASS-21 scores (Panel **A**) and scores on the depression (**B**), anxiety (**C**), and stress subscales (**D**) across the 12-week intervention (Tests 1–3) and at follow-up (Test 4) in Sugar (*n* = 27), Diet (*n* = 28) and Water groups (*n* = 25).

**Figure 4 nutrients-15-02191-f004:**
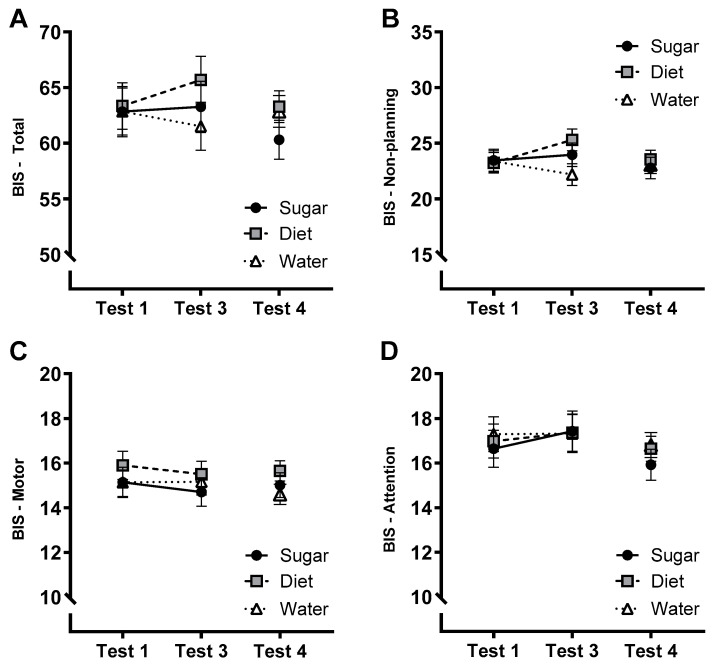
Estimated marginal means (±SE) of total BIS-11 scores (Panel **A**) and scores on the non-planning (**B**), motor (**C**), and attention subscales (**D**) across the 12-week intervention (Tests 1–3) and at follow-up (Test 4) in Sugar (*n* = 27), Diet (*n* = 28) and Water groups (*n* = 25).

**Figure 5 nutrients-15-02191-f005:**
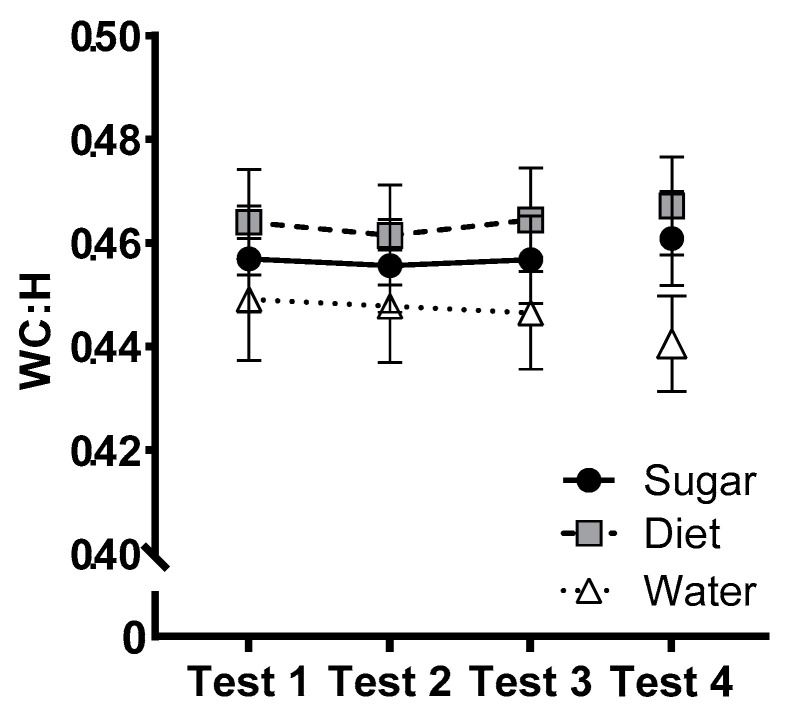
Estimated marginal means (±SE) for waist circumference to height ratio (WC:H) across the 12-week intervention (Test 1–Test 3) and at follow-up (Test 4) for Sugar (*n* = 27), Diet (*n* = 28) and Water (*n* = 25) groups. At follow-up, *n* = 25 (Sugar), 27 (Diet), 24 (Water).

**Figure 6 nutrients-15-02191-f006:**
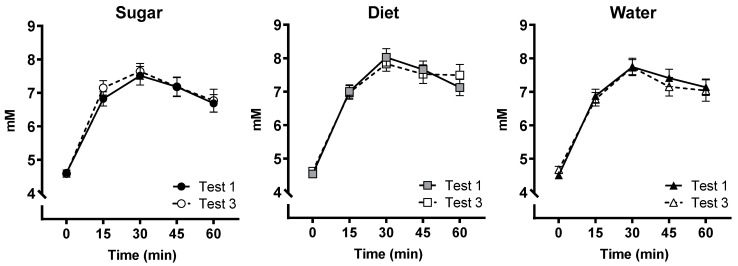
Blood glucose (mM, ±SE) was assessed in a modified 1-h oral glucose tolerance test at baseline (Test 1) and at the end of the 12-week intervention (Test 3) in Sugar (**left**; *n* = 27), Diet (**middle**; *n* = 28) and Water groups (**right**; *n* = 25), with no significant differences in the change from Test 1 to Test 3 across groups (non-significant test × time × group interaction).

**Figure 7 nutrients-15-02191-f007:**
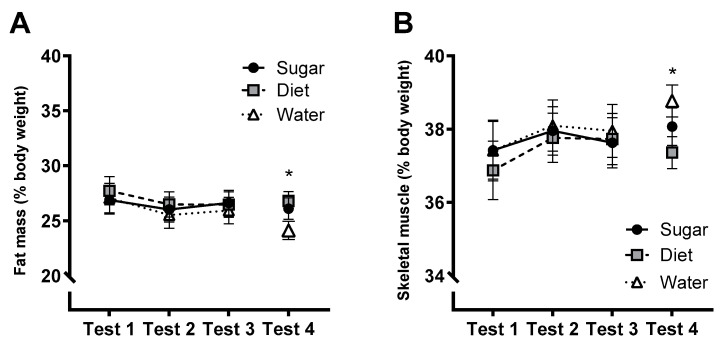
Estimated marginal means (±SE) of body fat (**A**) and skeletal muscle (**B**), measured via bioelectrical impedance and expressed as a percentage of body weight, across the 12-week intervention (Tests 1–3) and at follow-up (Test 4) for Sugar (*n* = 27), Diet (*n* = 28) and Water (*n* = 25) groups. * *p* < 0.05; Diet vs. Water groups (post-hoc pairwise comparison).

**Figure 8 nutrients-15-02191-f008:**
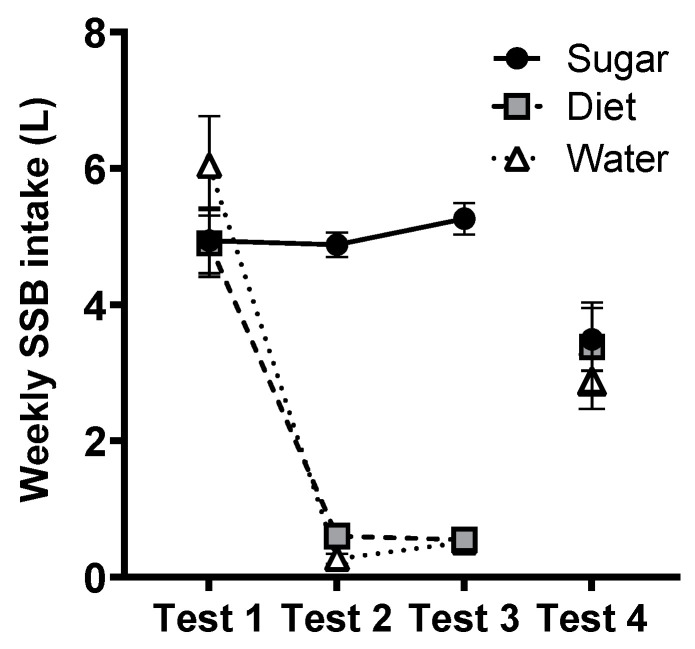
Estimated marginal means of average SSB intake (L/week ± SE, assessed via BEVQ) at baseline (Test 1), weeks 6 and 12 of the intervention (Tests 2 and 3), and at follow-up, 12 weeks after the intervention (Test 4) for Sugar (*n* = 27), Diet (*n* = 28) and Water (*n* = 25) groups.

**Figure 9 nutrients-15-02191-f009:**
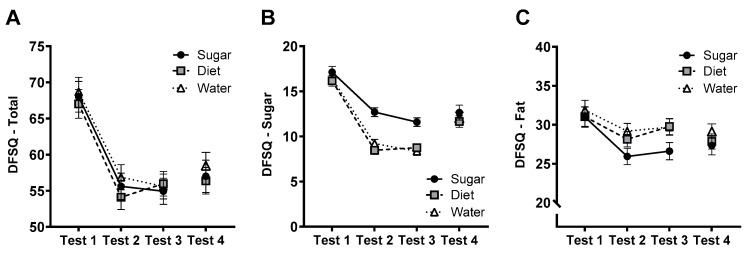
Estimated marginal means (±SE) of total DFSQ scores (**A**) and scores on the sugar (**B**) and fat (**C**) subscales across the 12-week intervention (Tests 1–3) and at follow-up (Test 4) for Sugar (*n* = 27), Diet (*n* = 28) and Water (*n* = 25) groups.

**Figure 10 nutrients-15-02191-f010:**
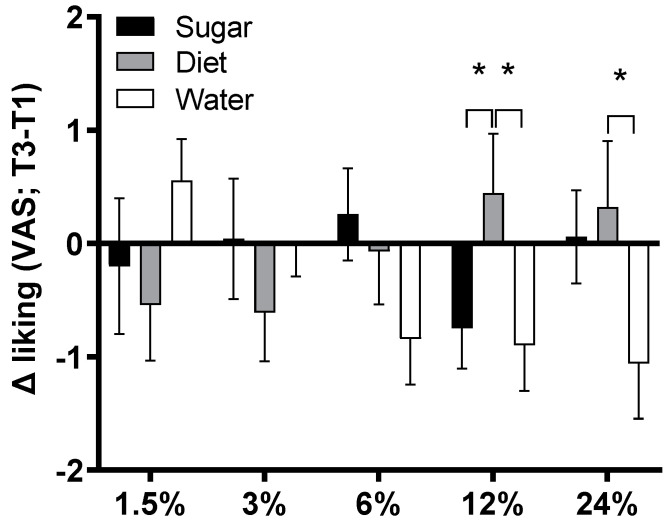
Average change liking ratings (by VAS; ±SE) for each sucrose concentration from baseline (Test 1) to the end of the 12-week intervention (Test 3) in Sugar (*n* = 27), Diet (*n* = 28), and Water (*n* = 25) groups. Participants rated their liking as 1.5%, 3%, 6%, 12%, and 24% sucrose solution on a 14cm VAS (14 = ‘liked very much’, 0 = ‘disliked very much’). A significant 3-way interaction (group × test × concentration) appeared to be driven by reductions in liking for 12% sucrose by Water and Sugar groups relative to the Diet group, and a reduction in liking for 24% sucrose in the Water group relative to the Diet group (* *p* < 0.05; post-hoc pairwise comparison).

## Data Availability

Data are available from the corresponding authors on reasonable request.

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
