# Peer review of "Switching from Sugar- to Artificially-Sweetened Beverages: A 12-Week Trial"

_nutrients, 2023, doi:10.3390/nu15092191_

Round 1

Reviewer 1 Report

This study used an unblinded 3-group parallel design to assess the effects of a 12-week intervention in which young healthy adults who regularly consumed SSBs were instructed to replace SSB intake with artificially-sweetened beverages or water, or to continue SSB intake. It has a great design and results. However, few comments must be addressed prior the acceptance for publication.

-Abstract: To separate aim, methods and results.

-Introduction: the explanation regarding to appetite modulation is appreciate.

-Methods: ok

-Results:  What is habitual food intake?

-Discussion: What are limitations of study? Parallel design?

Author Response

*We thank the reviewers for their constructive feedback, which we believe has improved the manuscript.

Best regards,

Mike Kendig (on behalf of all authors)

Reviewer 1

This study used an unblinded 3-group parallel design to assess the effects of a 12-week intervention in which young healthy adults who regularly consumed SSBs were instructed to replace SSB intake with artificially-sweetened beverages or water, or to continue SSB intake. It has a great design and results. However, few comments must be addressed prior the acceptance for publication.

-Abstract: To separate aim, methods and results.

*We have separated the Abstract into sub-headings

-Introduction: the explanation regarding to appetite modulation is appreciate.

*Thank you.

-Results:  What is habitual food intake?

*Habitual food intake refers to participants' standard day-to-day diet. We assessed one aspect of habitual diet (fat and sugar intake) via the DFS.

-Discussion: What are limitations of study? Parallel design?

*The main limitations of this study are that it was underpowered to detect significant differences based on higher than anticipated attrition, as covered in the Discussed.

Reviewer 2 Report

This is fairly straightforward trial, using relatively standard methods. It appears to have been planned and executed in line with current standards, although (as authors acknowledge) the strength of conclusions are limited by the size and power. My main concerns relate to the writing and data presentation, which unfortunately tend to obscure rather than facilitate reader understanding and interpretation of the context and main results. In several places the authors can better orient readers to the wider context of this research (totality of evidence) by reference to  systematic reviews, rather than individual studies. The overall flow of the sections would be more logical it were better aligned to the primary outcomes and main results for this. The results section in particular needs substantial re-drafting to improve clarity and readability. Authors need to put clear emphasis on conveying the results of main, pre-planned (hypothesis-led) analyses of primary interest, e.g. especially the overall group X time [week 3 vs 1] analyses. Secondary / post-hoc analyses should only follow these, regardless of what was or was not “statistically significant”. Those secondary analyses tend to be more distracting than informative or meaningful, and could perhaps be moved to supplementary material with only a brief mention in the main text. Reporting in many places of  F- values rather than p-values or actual differences (mean / 95% CI as an indication of effect sizes) is rather unhelpful. Lastly, I have some concerns about reliance on post-hoc analyses of within-group change-from-baseline data, rather than between-groups comparisons. This can be highly misleading and is inappropriate for drawing conclusions from  controlled trials (see e.g. Bland & Altman https://doi.org/10.1186/1745-6215-12-264; Vorland et al.,  https://doi.org/10.1038/s41366-021-00909-z).

Specific comments:

Line 14: Add the mean [SD or range] age and BMI

Since the primary (powered) outcome of this trial was cognitive measures, suggest to order the introduction in a way that places this background text (currently lines 55-61, perhaps expanded) first, and context for secondary outcomes following that. This cognitive measure seem also the most unique aspect of the trial, as there are many other, larger trials that have reported on body weight or metabolic outcomes of (reduced-)SSB interventions.

Paragraph starting line 40: Suggest to cite/summarise results from meta-analyses rather than a just a couple of randomly-chosen trials. There are also others that had a design similar to the present study (e.g. Piernas et al 2013 doi: 10.3945/ajcn.112.048405).

Lines 52-54 – This slightly misrepresents the actual results of Ebbeling et al (2020). Their analysis showed a significant overall between-group difference on preferred sweetness concentration, but no statistically significant differences in post-hoc comparisons of change between individual treatments, which they reported decreased from baseline within both water and artificially-sweetened beverage groups. Again, it would be fair to cite the relevant results from the (only) systematic review of this question around sweetness exposure and preferences (Appleton et al 2018 https://doi.org/10.1093/ajcn/nqx031) and/or additional trials since then.

Lines 61-70: It would be useful as a transition to the specific trial purpose and design, to have a little more justification for the specific cognitive domains of interest and likely duration needed to affect these, perhaps based on suggested plausible mechanisms. For example, if this secondary to effects on body weight or glycaemic control, then one probably needs a longer trial and change in these other parameters to have any likely impact.

Section 2.2: Recruitment and power is part of methods, but the description of the dropouts and study (analysis) population belongs in results. Authors can briefly summarise the flow from recruitment to analysis populations in text but are strongly advised to replace or supplement this with a detailed Consolidated Standards of Reporting Trials (CONSORT) subject flow diagram (in main body or supplementary material).

Line 81: A BMI of 17.5 is below the WHO cut-off for healthy weight (18.5). Please indicate the number of subjects who were underweight by this criterion.

Lines 86-87: Add mean [SD or range] BMI

Line 101: If possible, indicate the range of specific sweeteners present in the artificially-sweetened beverages.

Line 110: Authors are commended for clarity on the primary cognitive outcome measure, in agreement with the registered protocol. It seems less obvious to _also_ have had WC:H as a “primary” outcome, when the study was not also powered for this (or was it??).  

Section 2.5: Please add more information on blinding of the personnel with subject contact / administering the tests and measurements.

Section 2.6: Please add details on blinding of treatment codes (insofar as possible) during the experiment and per-protocol review and analyses (i.e. for whom and when were data locked and unblinded?).

Results, general:

The results text needs to be substantially improved to aid reader interpretation. It should start with a section on the study/analysis population (and CONSORT diagram) , where it may be helpful to also have a table describing the baseline characteristics (sex, age, BMI) of each group. The logical flow of the rest of the results would be better if all the cognitive tests are together, followed by anthropometric, then metabolic, then taste. This also aligns better to the stated aims of the trial.

The reporting for each outcome should preferably start with an explicit statement with exact p values for the main ANOVAs, with particular emphasis on the test 1-3 (weeks 0-12) time X treatment analyses [“group x test” in authors’ words], which is really the main analysis of interest. Instead, the key results for primary, pre-planned analyses are often lost amongst a large number of analyses of secondary interest, and many results also stated as F-values alone, which may be correct but few readers can readily interpret. For results of formal hypothesis testing, p-values are preferred. For expressing magnitude of effects, mean and 95% CI is preferred.

Figure 1B: Not sure this is needed.

Figure 2B: The figure is not needed, but suggest a scale range of -0.1 to +0.1 would in any case better represent the actual (trivial) differences.  

Figure 4: “…no significant differences observed” – Please clarify what analysis this refers to. Presumably this means “…no significant time x treatment effect” or some other analysis reflecting comparison of between-group changes test 3 vs test 1.

Lines 316-320 – The post-hoc within-group (change vs baseline) per-timepoint analyses here are potentially misleading (see general comments) and should be replaced by appropriately adjusted between-groups (difference in change) per-timepoint analyses. This may or may not alter the conclusions.

Lines 328-355, 369-378: As elsewhere, please reorder text in these sections to start with a clear, explicit statement of the results of the main pre-planned analyses (e.g. treatment x time). Any additional, secondary / post-hoc analyses and results should follow this. It’s unclear whether there were any pre-planned analysis or hypotheses relating to sex, and (if body weight is already included) it’s hard to see why this would be relevant. The reporting here risks looking like exploratory data (over-)analysis, and one may question the power and likely replicability of these results.  The readability of the paper may be substantially increased if authors limit text to a brief text description of such post-hoc / exploratory results, and put the detailed data and quantitative outcomes for these in supplementary material.

Lines 386-87, “Contrary to predictions, the study did not reveal improvements…”:  Suggest to put this in neutral terms, which also reflects the fact that all testing was 2-sided (i.e. tested for relative changes in _either_ direction). E.g. “There were no significant _differences_ between groups in…”.

Lines 390-92, and paragraph starting line 469: See previous comments. Any statements comparing groups should be drawn mainly from results of between-groups analyses, not within-group change from baseline. As noted, it would be fair to place this in the context of other research, not just Ebbeling et al (2020). The statement in line 479-482 is also incorrect, as Ebbeling et al (2020) reported significant within-group reductions in preferred sweetener concentration in both their water and artificially-sweetened beverage groups (their figure 3).

Lines 419 onwards: Discussion of the primary outcome(s), especially cognitive measures, should precede the discussion of dietary changes. The latter may be seen as evidence of compliance or a limitation or confounder, but was not the primary aim of the trial.

Author Response

This is fairly straightforward trial, using relatively standard methods. It appears to have been planned and executed in line with current standards, although (as authors acknowledge) the strength of conclusions are limited by the size and power. My main concerns relate to the writing and data presentation, which unfortunately tend to obscure rather than facilitate reader understanding and interpretation of the context and main results.

In several places the authors can better orient readers to the wider context of this research (totality of evidence) by reference to systematic reviews, rather than individual studies.

* We have added references to systematic reviews on the effects of SSB intake and specify where in response to the specific comments provided below. We are grateful for the reviewer’s assistance in this regard.

The overall flow of the sections would be more logical it were better aligned to the primary outcomes and main results for this. The results section in particular needs substantial re-drafting to improve clarity and readability. Authors need to put clear emphasis on conveying the results of main, pre-planned (hypothesis-led) analyses of primary interest, e.g. especially the overall group X time [week 3 vs 1] analyses. Secondary / post-hoc analyses should only follow these, regardless of what was or was not “statistically significant”. Those secondary analyses tend to be more distracting than informative or meaningful, and could perhaps be moved to supplementary material with only a brief mention in the main text. Reporting in many places of  F- values rather than p-values or actual differences (mean / 95% CI as an indication of effect sizes) is rather unhelpful.

*We have simplified the results so as to prioritise the primary analyses and omit more spurious interactions that were not of primary interest, as described in response to the specific comments listed below.

Lastly, I have some concerns about reliance on post-hoc analyses of within-group change-from-baseline data, rather than between-groups comparisons. This can be highly misleading and is inappropriate for drawing conclusions from controlled trials (see e.g. Bland & Altman https://doi.org/10.1186/1745-6215-12-264; Vorland et al.,  https://doi.org/10.1038/s41366-021-00909-z).

*We thank the reviewer for raising this. Within-group changes in sweetness preference appeared to drive the significant 3-way interaction observed in our primary analysis. Nonetheless, we recognise this limits the conclusions that can be drawn from trial data, and have replaced this with a new analysis of between-group differences in change scores. This shows significant differences in the change in liking ratings for 12% sucrose (Diet group increase liking more than Water and Sugar groups) and 24% sucrose (Diet increase liking more than Water group).

Specific comments:

Line 14: Add the mean [SD or range] age and BMI

*This information has been added.

Since the primary (powered) outcome of this trial was cognitive measures, suggest to order the introduction in a way that places this background text (currently lines 55-61, perhaps expanded) first, and context for secondary outcomes following that. This cognitive measure seem also the most unique aspect of the trial, as there are many other, larger trials that have reported on body weight or metabolic outcomes of (reduced-)SSB interventions.

*In the revised introduction the discussion of cognitive effects has been expanded to cite relevant work in humans and now precedes discussion of the metabolic effects, which are now also fleshed out with reference to relevant systematic reviews on SSB intake. Thanks for this suggestion.

Paragraph starting line 40: Suggest to cite/summarise results from meta-analyses rather than a just a couple of randomly-chosen trials. There are also others that had a design similar to the present study (e.g. Piernas et al 2013 doi: 10.3945/ajcn.112.048405).

*We have revised this paragraph and others in the Introduction to focus more on relevant meta-analyses and systematic reviews on the effects of SSBs and interventions to reduce consumption.

Lines 52-54 – This slightly misrepresents the actual results of Ebbeling et al (2020). Their analysis showed a significant overall between-group difference on preferred sweetness concentration, but no statistically significant differences in post-hoc comparisons of change between individual treatments, which they reported decreased from baseline within both water and artificially-sweetened beverage groups. Again, it would be fair to cite the relevant results from the (only) systematic review of this question around sweetness exposure and preferences (Appleton et al 2018 https://doi.org/10.1093/ajcn/nqx031) and/or additional trials since then.

* We have revised our description of the Ebbeling et al. (2020) study and now discuss it alongside the Appleton et al. systematic review.

Lines 61-70: It would be useful as a transition to the specific trial purpose and design, to have a little more justification for the specific cognitive domains of interest and likely duration needed to affect these, perhaps based on suggested plausible mechanisms. For example, if this secondary to effects on body weight or glycaemic control, then one probably needs a longer trial and change in these other parameters to have any likely impact.

*Thank you for this suggestion. We have expanded our discussion of the rodent and human literature on SSB intake (and SSB replacement), highlighting that cognitive effects are not dependent on body weight or adiposity changes.

Section 2.2: Recruitment and power is part of methods, but the description of the dropouts and study (analysis) population belongs in results. Authors can briefly summarise the flow from recruitment to analysis populations in text but are strongly advised to replace or supplement this with a detailed Consolidated Standards of Reporting Trials (CONSORT) subject flow diagram (in main body or supplementary material).

*We have moved the description of dropouts and the cohort details to the beginning of the results, and now include a CONSORT flow chart showing recruitment and attrition as Supplementary Figure 1.

Line 81: A BMI of 17.5 is below the WHO cut-off for healthy weight (18.5). Please indicate the number of subjects who were underweight by this criterion.

*This information has been added (4 participants, present in all 3 groups).

Lines 86-87: Add mean [SD or range] BMI

*This information has been added.

Line 101: If possible, indicate the range of specific sweeteners present in the artificially-sweetened beverages.

*This information has been provided (the sweeteners included aspartame, acesulfame-K and sucralose).

Line 110: Authors are commended for clarity on the primary cognitive outcome measure, in agreement with the registered protocol. It seems less obvious to _also_ have had WC:H as a “primary” outcome, when the study was not also powered for this (or was it??).  

*Our study was powered to detect effects of the intervention on the primary cognitive outcome. We selected changes in WC:H as our primary metabolic outcome variable but the study was not explicitly powered with this in mind. This has been clarified in the section detailing the primary outcomes.

Section 2.5: Please add more information on blinding of the personnel with subject contact / administering the tests and measurements.

*We have clarified that although the test administrator could not be blinded, cross-scoring for the primary outcome was done by team members blind with respect to intervention group, as follows: “Test sessions were recorded and scored live by the experimenter. Although blinding was not possible at this stage, audio recordings were not identifiable by treatment group, allowing for half the data to be re-scored at a later date by other team members not involved in test administration who were unaware of intervention group.”

Section 2.6: Please add details on blinding of treatment codes (insofar as possible) during the experiment and per-protocol review and analyses (i.e. for whom and when were data locked and unblinded?).

*We have clarified that blinding was not possible during data collection, but that raw data were entered into a master spreadsheet with treatment intervention coded until completion of the per-protocol review and statistical analyses.

Results, general:

The results text needs to be substantially improved to aid reader interpretation. It should start with a section on the study/analysis population (and CONSORT diagram) , where it may be helpful to also have a table describing the baseline characteristics (sex, age, BMI) of each group. The logical flow of the rest of the results would be better if all the cognitive tests are together, followed by anthropometric, then metabolic, then taste. This also aligns better to the stated aims of the trial.

*We have reorganised the Results as suggested so that the cognitive and affective tests are followed by the anthropometric, metabolic and taste preference data. We now also provide a CONSORT-style diagram as Supplementary Figure 1.

The reporting for each outcome should preferably start with an explicit statement with exact p values for the main ANOVAs, with particular emphasis on the test 1-3 (weeks 0-12) time X treatment analyses [“group x test” in authors’ words], which is really the main analysis of interest. Instead, the key results for primary, pre-planned analyses are often lost amongst a large number of analyses of secondary interest, and many results also stated as F-values alone, which may be correct but few readers can readily interpret. For results of formal hypothesis testing, p-values are preferred. For expressing magnitude of effects, mean and 95% CI is preferred.

*We have substantially cut down the Results section in order to prioritise the reporting of the primary interaction of interest. We have also removed reporting of more obscure interactions that were not the main focus. We now also report p-values and flag exploratory follow-up pairwise comparisons. In the few instances where these were significant we provide mean difference and the 95% CI.

Figure 1B: Not sure this is needed.

*This panel has been removed.

Figure 2B: The figure is not needed, but suggest a scale range of -0.1 to +0.1 would in any case better represent the actual (trivial) differences. 

*This panel has also been removed.

Figure 4: “…no significant differences observed” – Please clarify what analysis this refers to. Presumably this means “…no significant time x treatment effect” or some other analysis reflecting comparison of between-group changes test 3 vs test 1.

*We have revised this Figure legend to confirm that this refers to a non-significant interaction between group, test and time.

Lines 316-320 – The post-hoc within-group (change vs baseline) per-timepoint analyses here are potentially misleading (see general comments) and should be replaced by appropriately adjusted between-groups (difference in change) per-timepoint analyses. This may or may not alter the conclusions.

*As mentioned above, we now report an analysis of change scores in liking ratings and flag that post-hoc follow-up comparisons are exploratory. This analysis suggests that the Water group exhibited a greater reduction in liking for 12% and 24% sucrose relative to the Diet group, while the Sugar group also exhibited a significant reduction for 12% sucrose relative to the Diet group. We have also presented the data as change scores to match the analysis.

Lines 328-355, 369-378: As elsewhere, please reorder text in these sections to start with a clear, explicit statement of the results of the main pre-planned analyses (e.g. treatment x time). Any additional, secondary / post-hoc analyses and results should follow this. It’s unclear whether there were any pre-planned analysis or hypotheses relating to sex, and (if body weight is already included) it’s hard to see why this would be relevant. The reporting here risks looking like exploratory data (over-)analysis, and one may question the power and likely replicability of these results.  The readability of the paper may be substantially increased if authors limit text to a brief text description of such post-hoc / exploratory results, and put the detailed data and quantitative outcomes for these in supplementary material.

*As mentioned above, we have substantially condensed the results so that the group x test interaction is reported first, followed by any additional analyses (though many are now removed).

Lines 386-87, “Contrary to predictions, the study did not reveal improvements…”:  Suggest to put this in neutral terms, which also reflects the fact that all testing was 2-sided (i.e. tested for relative changes in _either_ direction). E.g. “There were no significant _differences_ between groups in…”.

*Thank you for this suggestion, now included.

Lines 390-92, and paragraph starting line 469: See previous comments. Any statements comparing groups should be drawn mainly from results of between-groups analyses, not within-group change from baseline. As noted, it would be fair to place this in the context of other research, not just Ebbeling et al (2020). The statement in line 479-482 is also incorrect, as Ebbeling et al (2020) reported significant within-group reductions in preferred sweetener concentration in both their water and artificially-sweetened beverage groups (their figure 3).

*We have revised our discussion in light of the new analyses of sweetness preference, which show a more complex set of changes, and to convey that the differences in sweet taste preference reported by Ebbeling et al. (2020) were within-group.

Lines 419 onwards: Discussion of the primary outcome(s), especially cognitive measures, should precede the discussion of dietary changes. The latter may be seen as evidence of compliance or a limitation or confounder, but was not the primary aim of the trial.

*The two paragraphs discussing the cognitive results now precede the discussion of dietary changes; thank you.

Round 2

Reviewer 2 Report

The authors have satisfactorily addressed my comments. The revised manuscript is substantially improved in clarity and readability, and better conveys the context and key outcomes.